# The Impact of Venous Invasion on the Postoperative Recurrence of pT1–3N0cM0 Gastric Cancer

**DOI:** 10.3390/jpm13050734

**Published:** 2023-04-26

**Authors:** Yasuo Imai, Yoshihiro Kurata, Masanori Ichinose

**Affiliations:** 1Department of Diagnostic Pathology, Ota Memorial Hospital, SUBARU Health Insurance Society, 455-1 Oshima, Ota City 373-8585, Gunma, Japan; 2Department of Digestive Surgery, Shioya Hospital, International University of Health and Welfare, Yaita City 329-2145, Tochigi, Japan; kuratayoshihiro@chiba-u.jp (Y.K.); surg-ichinose@iuhw.ac.jp (M.I.); 3Department of Frontier Surgery, Graduate School of Medicine, Chiba University, Chiba City 263-8522, Chiba, Japan

**Keywords:** gastric cancer, stomach cancer, N0, M0, venous invasion, recurrence, metastasis, adjuvant chemotherapy, EVG

## Abstract

The impact of venous invasion (VI) on postoperative recurrence in pathological (p)T1–3N0 clinical (c)M0 gastric cancer (GC) remains unclear. We investigated the association of VI grade with prognosis in 94 (78 stage I and 16 stage IIA) patients. VI was graded during pathological examinations based on the number of VIs per glass slide as follows: v0, 0; v1, 1–3; v2, 4–6; and v3, ≥7. Filling-type invasion in veins with a minor axis of ≥1 mm increased VI grade by 1. Four (4.3%) patients experienced recurrence. Recurrence increased with pT (pT1, 0.0%; pT2, 11.1%; pT3, 18.8%) and VI grade (v0, 0.0%; v1, 3.7%, v2, 14.3%; and v3, 40.0%). Recurrence was significantly more frequent in pT3 than pT1 and in v2 + v3 than v0 (*p* = 0.006 and 0.005, respectively). Kaplan–Meier curve analyses demonstrated a significant decrease in recurrence-free survival according to pT (*p* = 0.0021) and VI grade (*p* < 0.0001). Multivariate Cox analysis revealed a significant association of VI grade with recurrence (*p* = 0.049). These results suggest that VI grade is a potential recurrence predictor for pT1–3N0cM0 GC. No recurrence can be expected in cases with pT1 or VI grade v0. Adjuvant therapy might be considered for pT3 or VI grade v2 + v3.

## 1. Introduction

Gastric cancer (GC) is one of the most common and deadly neoplasms in the world. In 2020, GC was responsible for an estimated 1,089,103 new cases and 768,793 deaths globally, ranking fifth for incidence and fourth for mortality [1]. The 5-year relative survival rates for GC are 70% for the localized stage, 32% for the regional stage, 6% for the distant stage, and 32% for all stages combined [2]. Thus, the GC prognosis is poor when it spreads outside of the stomach.

Pathologists routinely evaluate the histological subtype, grade of differentiation, depth of tumor invasion (pT), lymphovascular invasion, nodal metastasis (pN), and resection margin status (R) of resected GC specimens. Clinicians determine the postoperative treatment strategy, such as whether to perform adjuvant chemotherapy (AC), based on the pathology report. AC is performed to decrease the risk of recurrence and/or metastasis. In the United States, surveillance or adjuvant fluoropyrimidine is recommended for pT2N0 GC patients and adjuvant fluoropyrimidine plus oxaloplatin is recommended for pT3N0 GC patients according to the National Comprehensive Cancer Network Guidelines [3]. Adjuvant fluoropyrimidine-based chemoradiotherapy is recommended for specific cases, such as R1 and R2 resections and less than a D2 nodal dissection [3]. In Japan, AC with S-1 (tegafur/gimeracil/oteracil) has been recommended for stages II/III GC patients except for those with T1N2/N3 and T3N0 based on the Adjuvant Chemotherapy Trial of S-1 for Gastric Cancer (ACTS-GC) [4].

Metastasis is caused by tumor cell spread via lymphatic, vein, or dissemination; venous invasion (VI) is theoretically a risk factor for hematogenous metastasis. Cancer cells that invade the veins in the stomach reach the liver via the portal system, reach the lungs from the liver via the inferior vena cava, and finally enter the systemic circulation system. In the Union for International Cancer Control (UICC) tumor, node, metastasis (TNM) staging system (8th ed.), V1 and V2 are defined as microscopic and macroscopic VI, respectively; however, VI is not implicated in UICC TNM staging system stage definitions [5]. Meanwhile, the Japanese Classification of Gastric Carcinoma (JCGC) (15th ed.) classifies VI as V0 (none), V1a (mild), V1b (moderate), and V1c (severe) based on the pathologist’s subjectivity [6]. However, there are no such objective criteria for grading VI as serve for prognostic stratification. The purpose of this study was to determine the criteria in surgically resected GC without nodal or distant metastasis at the time of surgery; patients with these GC types may be subject only to surveillance as a low-risk group for postoperative recurrence in Japan.

## 2. Materials and Methods

### 2.1. Patients

This study analyzed 234 consecutive patients who underwent resection of primary GC at the International University of Health and Welfare, Shioya Hospital between 2006 and 2019. Patients with gastroesophageal junction cancer (Siewert type II) [7], carcinoma in situ/high-grade dysplasia, squamous cell carcinoma, and distant metastasis found prior to or at the time of surgery (clinical (c) or pathological (p) M1), and patients without nodal dissection were excluded. Patients with any invasive cancers (GC and cancers other than GC) that were resected between 5 years before gastrectomy and 5 years after gastrectomy were excluded, except for those with synchronous multiple GCs at gastrectomy. On the other hand, patients with asynchronous invasive cancers (independently developed GC and cancers other than GC) that occurred later than 5 years after gastrectomy were included and censored at the time of diagnosis of the new tumors. Clinicopathologic information was retrieved via the electronic chart system; patients without complete clinical information were excluded. After pathological examination of the resected stomach and dissected lymph nodes, patients with GC invading up to the subserosal layer (pT1–3) without nodal metastasis (pN0) were analyzed in this study.

AC was initiated postoperatively in 1 patient with pStages I and 8 patients with pStages IIA within 4 to 6 weeks postoperatively at the clinicians’ discretion (Appendix A). The regimens were principally either S-1 or paclitaxel in the case of S-1 intolerance for one year [4,8]. Schedules and doses were modified according to the patient’s performance status. Patients were followed up every 3 weeks during AC. Patients who did not receive AC were followed up monthly for the first 1 to 2 years according to the patients’ pTNM stages. Follow-up was then continued every 2 months until 5 years postoperatively or censored for social reasons. Blood tests were performed every 2 months, and gastroscopy and contrast-enhanced computed tomography were performed every 6 months for the first year and yearly for 4 more years. No patients received adjuvant radiotherapy or chemoradiotherapy.

### 2.2. Pathological Examination

All surgical specimens were routinely processed for pathological diagnosis. Early cancers extending no further than the submucosa were subjected to microscopic inspection of the whole tumor area. In cases of advanced cancer invading no less than proper muscle, the maximal cut surface of the tumor involving the transition between the tumor and normal mucosa and the cut surface involving the deepest tumor penetration were microscopically inspected. Clinicopathologic classifications and stage groupings were performed based on the World Health Organization (WHO) classification of tumors of the stomach (5th ed.) and the UICC TNM staging system (8th ed.) [5,9]. Histological classifications were based on the predominant histologic pattern of the carcinoma [10,11] and cancer stromal volumes and infiltration patterns were classified based on the JCGC (3rd English ed.) [12]. Elastica van Gieson (EVG) staining was performed in two or more, if necessary, sections that included the deepest tumor penetration and the area of transition between the tumor and normal mucosa. Each section usually contained approximately 2 to 5 cm^2^ of tissue per glass slide. When tumor cells invaded or were located in the tubular structure formed by an elastic plate adjacent to the artery (i.e., an adventitia of the vein), VI was diagnosed (Appendix A). VI in each section was graded according to the number of VIs irrespective of the location: v0, no venous invasion; v1, 1–3 venous invasions per glass slide; v2, 4–6 venous invasions per glass slide; v3, ≥7 venous invasions per glass slide. Filling-type VI, in which tumor cells filled the macroscopically identifiable vein with a minor axis of ≥1 mm, increased the VI grade by 1. The VI grade in each case was based on the maximal grade in the investigated sections. The most predominant histological subtypes, the deepest tumor invasion, the highest VI grade, and the highest pTNM stage were recorded when patients had multiple synchronous GCs.

### 2.3. Statistical Analysis

Categorical parameters between two patient cohorts were compared using Fisher’s exact test for 2 × 2 cross-tabulations or the chi-square test with or without Yates’ correction as appropriate for m × n cross-tabulations. Age was compared using the Mann–Whitney *U* test, and the number of dissected lymph nodes was compared using the unpaired Student’s *t* test. Associations between clinicopathologic parameters and recurrence/metastasis, which are collectively referred to as recurrence hereinafter, were analyzed using the univariate Cox regression analysis. Multivariate Cox regression analysis was performed using the forced entry method on selected parameters with *p* values < 0.05 using univariate analysis. Survival curves were analyzed using the Kaplan–Meier method with the log-rank test; *p* values < 0.05 were considered significant. Statistical analyses were performed using IBM SPSS Statistics 20 (IBM Corp., Armonk, NY, USA).

## 3. Results

### 3.1. Clinicopathologic Features of Patients with pT1–3N0cM0 GC and Associations of the VI Grade with Recurrence

During the study period, 234 patients underwent gastrectomy; however, 69 patients were excluded mostly because of multiple primary cancers between 5 years before gastrectomy and 5 years after gastrectomy. An additional 71 patients were excluded because they had lymph nodal metastasis. Finally, data for 94 patients (78 pStage I and 16 pStage IIA) were analyzed (Table 1). Patients included 65 males and 29 females, aged 42–92 (median: 70). Among them, 69 and 23 underwent distal gastrectomy and total gastrectomy, respectively. Laparoscopic gastrectomy was performed in 13 patients, none of whom had recurrence. Surgical details were available for 62 patients who underwent gastrectomy after 2012. Operation time was 264 ± 73 min, and the blood loss was 222 ± 245 mL. Blood transfusion was performed in five patients, one of whom later experienced recurrence. Splenectomy was not performed in any patient. There were no significant differences in surgical details between patients with recurrence and those without recurrence (data not shown). The follow-up periods from surgery to cancer-related death or censoring ranged from 20 to 4888 days (median: 2044 days). Recurrence was observed in four (4.3%) patients, one with pStage I and three with pStage IIA GCs. Recurrence sites included the residual stomach in one (1.1%), peritoneum in two (2.1%), and liver in two (2.1%) patients (Table 2). There were significant differences in pT, VI grade, and pTNM stage between patients with and without recurrence (Table 1). Although the frequency of recurrence in patients with pN0 GC was very low, the recurrence rate significantly increased according to pT as follows: pT1 (invasion to the lamina propria or submucosa) (0.0%), pT2 (invasion to the muscularis propria) (11.1%), and pT3 (invasion to the subserosa) (18.8%) (pT1 vs. pT3, *p* = 0.006, using Fisher’s exact test). The recurrence rate also significantly increased according to the VI grade as follows: v0 (0.0%), v1 (3.7%), v2 (14.3%), and v3 (40.0%) (v0 vs. v3, *p* = 0.006 and v0 (V0) vs. v2 + v3 (V1High (V1H)), *p* = 0.005, using Fisher’s exact test).

### 3.2. Recurrence-Free Survival Analysis Using the Kaplan–Meier Method

We next depicted the Kaplan–Meier curves of recurrence-free survival (RFS) of pT1–3N0cM0 GC patients according to pT and VI grades. There were significant differences in pT1, pT2, and pT3 RFS curves (*p* = 0.0021, Figure 1A); pT3 had a significantly worse RFS than pT1 (*p* = 0.0002). The RFS rate of pT1 patients was 100% even 10 years after surgery, whereas the RFS rate of pT3 patients fell to 81.3% one-year postoperatively. Furthermore, there were significant differences in RFS curves among v0–3 (*p* < 0.0001) and among v0 (V0), v1 (V1Low (V1L)) and v2 + v3 (V1H) (*p* = 0.0003) (Figure 1B); v2, v3, and v2 + v3 (V1H) patients had significantly worse RFSs (*p* = 0.0055, *p* < 0.0001 and *p* = 0.0001, respectively) than v0 (V0). The RFS rate of v0 (V0) was 100% even 10 years after surgery, whereas the RFS rates of v3 and v2 + v3 (V1H) fell to 60.0% and 75.0%, respectively, by one-year postoperatively.

### 3.3. Multivariate Cox Regression Analysis of the Recurrence Predictive Factor

Finally, multivariate Cox analysis was performed to adjust for the confounding factors of the recurrence predictive factors. pT, VI grade, pTNM stage, and AC were selected as candidate prognostic factors using the univariate analyses (Table 3). In this study, pT directly paralleled with the pTNM stage. AC was performed at the physician’s discretion considering the pathology report, including pT and VI grade. In fact, AC was positively associated with recurrence, although it is intended to prevent recurrence. Accordingly, pTNM stage and AC were excluded from the multivariate analysis, and we found that only VI grade was significantly associated with postoperative recurrence (*p* = 0.049).

## 4. Discussion

AC is performed to decrease the risk of recurrence. As GC patients without nodal or distant metastases (pN0cM0) can intrinsically expect favorable prognoses, AC is not intended to treat pT1–3N0cM0 GC in Japan based on the ACTS-GC trial [4]. However, as presented in this study, there are a small but certain number of patients who suffer from recurrence after surgical resection. AC indications and regimens to prevent postsurgical pN0 GC recurrence should, therefore, be considered. In fact, the SWOG INT-0116 trial [13] exhibited positive results for AC with fluorouracil plus leucovorin for stages IB–III GC, including T2–3N0, using the UICC TNM staging system (8th ed.). To predict pN0 GC postoperative recurrence, we focused on pathological VI, which is theoretically linked to hematogenous metastasis. In the UICC TNM staging system (8th ed.), V1 and V2 are defined as microscopic and macroscopic VI, respectively [5]. In contrast, the latest JCGC (15th ed.) classifies VI as V0 (none), V1a (mild), V1b (moderate), and V1c (severe) based on the pathologist’s subjectivity, without distinguishing between microscopic and macroscopic VI [6]. In this study, we created an objective VI-grade criterion by integrating the VI number per glass slide and the size of the invaded veins under the assumption that the VI number and size of the invaded veins would both positively correlate with metastatic potential.

This study’s most significant difficulty was recruiting a sufficient number of cases with an event, that is, postsurgical recurrence. Recent advances in endoscopy have made it easier to find early-stage GCs and many GCs invading no more than pT1b (submucosal invasion) can be resected endoscopically. As a result, the number of GCs that undergo surgical resection has been decreasing. Lymph nodes are likely to be involved in locally advanced GC that undergoes surgical resection. Furthermore, many of our patients were elderly; many patients developed invasive cancers in other organs and were excluded from the analysis. In fact, 234 patients underwent gastrectomy during the study period, but only 94 cases were eligible for this study. Furthermore, only four patients experienced recurrence because of the intrinsic excellent prognosis of pT1–3N0cM0 GC.

Despite the small number of events, our data clearly demonstrated that pT and VI grade were potential prognostic factors for recurrence. The recurrence rate was 0.0% (0/69) in pT1N0 GC, whereas it was 18.8% (3/16) in pT3N0 GC. This difference was statistically significant. RFS curve analysis using the Kaplan–Meier method also demonstrated significantly worse prognoses for pT3N0 than for pT1N0. Although the permissible recurrence rate may vary according to physicians, a recurrence rate as high as 20% may not be overlooked. Likewise, the recurrence rate was 0.0% (0/55) in GC with v0 (V0), whereas it was 25.0% (3/12) in GC with v2 + v3 (V1H) GC. This difference was also statistically significant. RFS curve analysis using the Kaplan–Meier method also demonstrated significantly worse prognoses for v3 and v2 + v3 (V1H) than for v0 (V0). The v3 recurrence rate (as high as 40%), and even that of v2 + v3 (V1H) (as high as 25%), may not be overlooked. Although multivariate Cox analysis failed to demonstrate that pT had a statistical significance as a prognostic factor, it revealed that VI grade was statistically significant as a prognostic factor. Given these results, we think that AC may be considered to treat GC with pT3 and at least VI grades v2 and v3 (V1H).

Presumably due to difficulties in collecting cases, we found only two studies similar to the present study. Yu et al. reported that 6 of 155 (3.9%) patients with pT2N0 GC developed recurrence after surgical resection [14]. The recurrence site was the liver in three cases, the peritoneum in two cases, and the stomach in one case. VI was not a significant prognostic factor of recurrence in their study; however, VI was evaluated using hematoxylin and eosin staining, which may have lower sensitivity than EVG staining. Araki et al. investigated VI’s prognostic impact in 130 GC patients at stage pT2N0 according to the UICC TNM staging system (8th ed.) [15]. Recurrences occurred in 12 (9.2%) patients; seven in the liver, two in the peritoneum, three in the lymph nodes, and one as a locoregional recurrence. Consistent with our results, they also found that moderate or marked venous invasion (v2 or v3) was the sole significant predictor of recurrence. In their study, VI was graded using EVG staining as follows: v0, VI was not found on any slide examined; v1, one or two VI sites among all eight slides examined; v2, intermediate level between v1 and v3; and v3, one or more VI sites on every slide examined. In addition, the pT2N0 GC recurrence rates in the above two studies were comparable to our results (1/9 cases, 11.1%).

VI has been reported to be associated with postoperative hematogenous metastasis [16,17]. In our study, two of four recurrences developed in the liver and two recurrences were in the peritoneum. Nakanishi et al. reported that VI was not associated with peritoneal recurrence, which was significantly associated with lymph node metastasis and differentiation grade [16]. Our two cases with peritoneal recurrence were poorly differentiated adenocarcinoma, non-solid type, in which poorly cohesive cancer cells diffusely infiltrated with a strong stromal reaction. Although histology was not demonstrated to be a significant predictor of recurrence in the present study, Nakanishi’s theory may apply to our cases with peritoneal recurrence.

According to the ACTS-GC study, S-1 monotherapy could significantly reduce nodal and peritoneal metastases, but could hardly control the hematogenous metastasis [4]. In fact, two of three patients with recurrences of v2 or v3 in this study were administered adjuvant S-1, but they suffered from liver metastasis. Recently, the CLASSIC study performed in East Asia reported that CapeOX (capecitabin plus oxaliplatin) could well control distant metastases after surgical resection of stages II/III GC, but could not control peritoneal and lymph nodal metastases [18]. The JACCRO GC-07 trial performed in Japan reported that postoperative S-1 plus docetaxel significantly decreased hematogenous and nodal metastases compared to S-1 alone after surgical resection of stage III GC [19]. These regimens may be candidates for AC when v2 or v3 is demonstrated in differentiated adenocarcinoma during the postoperative pathological examination. In addition, perioperative FLOT (fluorouracil, leucovorin, oxaliplatin, and docetaxel) has been reported to be superior to ECF therapy (epirubicin, cisplatin, and fluorouracil) and is now considered the new standard chemotherapy regimen for resectable GC in Europe [20]. Furthermore, adjuvant SOX (S-1 plus oxaliplatin) for stage II/III, node-positive GC has been reported to be superior to S-1 monotherapy in Korea [21]. Although these regimens’ site-specific inhibitory effects on recurrence have not been described, they may be options for AC in this situation.

Our study had some limitations. First, this study was retrospective and conducted at a single institution. Therefore, the number of events was very small. However, given the difficulty in recruiting eligible cases and the small number of similar studies available, we believe that our study will aid in determining treatment strategies for patients with pT1–3N0cM0 GC. Second, this study’s time span ranged over 14 years, during which surgical techniques substantially changed (e.g., spleen preservation and the introduction of laparoscopic surgery); this should be remembered when interpreting our data.

## 5. Conclusions

We investigated the impact of venous invasion in surgically resected pT1–3N0cM0 gastric cancer on recurrence. There was no recurrence in cases categorized as pT1 or no venous invasion. The grade of venous invasion and pT are potential recurrence predictors, and we think that pT3 gastric cancer patients and gastric cancer patients with grades v2 and v3 venous invasion could be considered for participation in future studies to clarify whether or not adjuvant chemotherapy should be recommended to these patients.

## Figures and Tables

**Figure 1 jpm-13-00734-f001:**
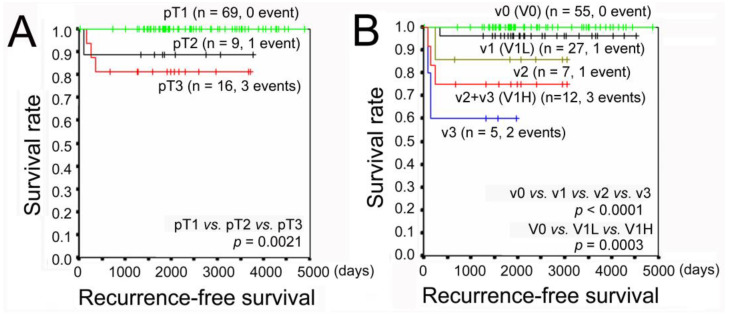
Recurrence-free survival of pN0 gastric cancer patients according to the depth of tumor invasion (**A**) and grade of venous invasion (**B**). v0, no venous invasion; v1, 1–3 invasions/slide; v2, 4–6 invasions/slide; v3, ≥7 invasions/slide. Filling-type of venous invasion in a macroscopically identifiable vein with a minor axis of ≥1 mm increased the VI grade by 1.

**Table 1 jpm-13-00734-t001:** Clinicopathologic characteristics of pT1–3N0cM0 gastric cancer.

Parameters		Total (*n* = 94)	Recurrence		*p* Value
			Yes (*n* = 4)	No (*n* = 90)	
Age					
	Median (range)	70 (42–92)	77 (70–86)	70 (42–92)	0.130
Sex					
	Male/Female	65/29	4/0	61/29	0.226
Location					
	Upper/Middle/Lower	21/16/57	0/0/4	21/16/53	0.980
Surgery					
	DG/TG/others *	69/23/2	3/1/0	66/22/2	0.956
Lymph node dissection					
	D1/D1+/D2	8/44/42	0/2/2	8/42/40	0.822
Number of dissected lymph nodes					
	Mean ± SD	26.6 ± 12.0	18.0 ± 8.0	27 ± 12.1	0.145
Synchronous multiple GCs					
	Yes/No	10/84	1/3	9/81	0.367
Histology					
	wel/mod/muc/por	33/23/0/38	2/0/0/2	31/23/0/36	0.501
Depth of tumor invasion					
	pT1/pT2/pT3	69/9/16	0/1/3	69/8/13	0.002
Cancer stromal volume					
	med/int/sci/not described *	3/43/11/37	0/2/2/0	3/41/9/37	0.659
Tumor infiltration pattern					
	INFa/INFb/INFc/not described *	3/46/19/26	0/2/2/0	3/44/17/26	0.925
VI grade					
	v0/v1/v2/v3	55/27/7/5	0/1/1/2	55/26/6/3	<0.001
Resection margin status					
	R0/R1 + R2	94/0	4/0	90/0	N.C.
pTNM					
	I/IIA	78/16	1/3	77/13	0.015
Neoadjuvant chemotherapy					
	Yes/No/unknown *	0/82/12	0/4/0	0/78/12	N.C.
Adjuvant chemotherapy					
	Yes/No/unknown *	9/73/12	2/2/0	7/71/12	0.058

* not included in the statistical analysis. DG, distal gastrectomy; TG, total gastrectomy; SD, standard deviation; GC, gastric cancer; wel, well-differentiated adenocarcinoma; mod, moderately differentiated adenocarcinoma; muc, mucinous adenocarcinoma; por, poorly differentiated adenocarcinoma, either non-solid type or solid type; VI, venous invasion; TNM, tumor node metastasis; N.C., not calculated; v0, no venous invasion; v1, 1–3 invasions/slide; v2, 4–6 invasions/slide; v3, ≥7 invasions/slide. Filling-type VI in a macroscopically identifiable vein with a minor axis of ≥1 mm increased the VI grade by 1.

**Table 2 jpm-13-00734-t002:** Patients who experienced postoperative recurrence.

Patients	No. 111	No. 115	No. 189	No. 197
Age	71	70	83	86
Sex	Male	Male	Male	Male
Location	Low	Low	Low	Low
Surgery	DG	TG	DG	DG
Lymph node dissection	D2	D1+	D2	D1+
Number of dissected lymph nodes	14	30	15	13
Synchronous multiple GCs (Number)	Yes (2)	No	No	No
Histology	por, non-solid	por, non-solid	wel	wel
Depth of tumor invasion	pT3	pT3	pT3	pT2
Cancer stromal volume	sci	sci	int	int
Tumor infiltration pattern	INFc	INFc	INFb	INFb
VI grade	v1	v3	v2	v3
Resection margin status	R0	R0	R0	R0
Neoadjuvant chemotherapy	No	No	No	No
Adjuvant chemotherapy	No	No	S-1	S-1
Site of recurrence	Residual stomach Peritoneum	Peritoneum	Liver	Liver

GC, gastric cancer; VI, venous invasion; DG, distal gastrectomy; TG, total gastrectomy; wel, well-differentiated adenocarcinoma; por, poorly differentiated adenocarcinoma; v1, 1–3 invasions/slide; v2, 4–6 invasions/slide; v3, ≥7 invasions/slide. Filling-type VI in a macroscopically identifiable vein with a minor axis of ≥1 mm increased the VI grade by 1.

**Table 3 jpm-13-00734-t003:** Recurrence predictors of pT1–3N0cM0 gastric cancer using multivariate Cox analysis.

Parameters		HR (95% CI)	*p* Value
Univariate analysis			
	Age	1.087 (0.969–1.219)	0.154
	Sex (Male)	35.79 (0.005–276 × 10^3^)	0.433
	Surgery (TG)	1.022 (0.106–9.829)	0.985
	Lymph node dissection (D2)	1.245 (0.175–8.841)	0.826
	Number of dissected lymph nodes	0.938 (0.859–1.024)	0.152
	Synchronous multiple GCs (Yes)	3.055 (0.318–29.372)	0.333
	pT	6.465 (1.283–32.591)	0.024
	Histology (por)	1.507 (0.212–10.698)	0.682
	Tumor infiltration pattern (INFc)	2.595 (0.366–18.426)	0.340
	Cancer stromal volume (sci)	4.290 (0.604–30.466)	0.145
	VI grade	4.936 (1.758–13.653)	0.002
	pTNM stage (IIA)	15.033 (1.563–144.603)	0.019
	Adjuvant chemotherapy (Yes)	8.972 (1.262–63.807)	0.028
Multivariate analysis			
	pT	3.472 (0.550–21.932)	0.186
	VI grade	3.099 (1.004–9.571)	0.049

HR, hazard ratio; CI, confidence interval; TG, total gastrectomy; GC, gastric cancer; pT, depth of tumor invasion; VI, venous invasion; pTNM, pathological tumor node metastasis.

## Data Availability

All data generated or analyzed in this study are included in this published article.

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
