# Peer review of "The Impact of Venous Invasion on the Postoperative Recurrence of pT1–3N0cM0 Gastric Cancer"

_jpm, 2023, doi:10.3390/jpm13050734_

Round 1

Reviewer 1 Report

This is an interesting topic and deserves further discussion.

The hypothesis and aims at the end of the introduction are confusing. The sentence starting with "The purpose of this study was to determine the criteria..", you never actually say what is being determined. 

The final sentence in the introduction belongs in the methods section. 

Methods: Why the 5 year cut off? Once the anatomy has been altered, it has been altered. 

I believe putative is being misused. 

Results/Discussion: In other countries, we consider periooperative chemotherapy in T2N0+. Would you consider backing down treatment in those without V1 and lower TNM stage. Would comment on the heterogeneity of treatment around the world. 

Author Response

(C1) The hypothesis and aims at the end of the introduction are confusing. The sentence starting with "The purpose of this study was to determine the criteria..", you never actually say what is being determined.

(A1) The principal aim of this study was to determine the objective criteria for grade of venous invasion (VI) in surgically resected gastric cancer (GC). The VI grading system hitherto proposed had been a subjective one based on pathologists’ subjectivity. In this study, we proposed the following VI grading system: v0, no venous invasion; v0, 1 to 3 venous invasions per glass slide; v2, 4 to 6 venous invasions per glass slide; v3, ³ 7 venous invasions per glass slide. Filling-type VI, in which tumor cells filled the macroscopically identifiable vein with a minor axis of ³ 1 mm increased the grade by 1. These sentences are described in the 2.2. Pathological Examination section. We then investigated the validity of this VI grading system for predicting prognosis. Later, this VI grading system was demonstrated to be a significant predictor of recurrence in node-negative GC by the multivariate Cox analysis. (Line 192) 

(C2) The final sentence in the introduction belongs in the methods section.

(A2) We modified the sentence just before the final sentence and deleted the final sentence according to the reviewer’s comment. (Lines 60-63)

(C3) Methods: Why the 5 year cut off? Once the anatomy has been altered, it has been altered.

(A3) We are sorry that we cannot well understand the reviewer’s comment. If you mean the reason for which patients with invasive cancers (other than gastric cancer that underwent gastrectomy) that were resected between 5 years before gastrectomy and 5 years after gastrectomy were excluded, it is because cancer patients who are disease-free for more than 5 years after therapy are usually thought to be cured of cancer. Therefore, we thought that recurrence of cancer in this study could be considered to be a recurrence of gastric cancer that underwent gastrectomy.

(C4) I believe putative is being misused.

(A4) “Putative” was deleted and the relevant sentences were revised. (Lines 94-99)

(C5) Results/Discussion: In other countries, we consider periooperative chemotherapy in T2N0+. Would you consider backing down treatment in those without V1 and lower TNM stage. Would comment on the heterogeneity of treatment around the world.

(A5) We do not intend to comment on the treatment performed in North America and Europe. In Japan, many gastric cancers undergo upfront surgery followed by adjuvant chemotherapy with S-1 if it was deemed necessary. To be noted is that pT2N0 and pT3N0 gastric cancer with curative resection are not indication of adjuvant chemotherapy based on the ACTS-GC trial and some of them suffer from recurrence. We therefore attempted to find out high-risk group of node-negative gastric cancer for recurrence which may require adjuvant therapy from the viewpoint of venous invasion. We also well recognize the heterogeneity of treatment around the world and referred to it in the text. (Lines 42-49 & 286-290)

Finally, we had our manuscript revised by the MDPI English editing service (Order number: English-64895) as per the reviewer’s suggestion.

Reviewer 2 Report

I would like to congratulate the authors of the paper. I have a few suggstions to make which in my opinion will improve the image of the manuscript.

1) In the introduction: You have to correct what the western guidelines propose for adjuvant treatment for gastric cancer. These guidelines are in favor of adjuvant chemotherapy and not chemo-radiotherapy which is suggested only in specific cases. I would also suggest to add that in western countries the suggestion for most patients is perioperative chemotherapy with FLOT.

2) You have written the follow up period in days. I would suggest to use months or years rather than days (page 4, line 148)

3) In the section "Conclusions" I think that the conclusion to consider adjuvant chemotherapy in patients with vascular invasion is not a result of your study. You might want to say: "these patients could be consider for participation in future studies of adjuvant chemotherapy".

Author Response

(C1) I would like to congratulate the authors of the paper. I have a few suggstions to make which in my opinion will improve the image of the manuscript.

(A1) We thank Reviewer 2 for his/her encouraging comment.

(C2) In the introduction: You have to correct what the western guidelines propose for adjuvant treatment for gastric cancer. These guidelines are in favor of adjuvant chemotherapy and not chemo-radiotherapy which is suggested only in specific cases. I would also suggest to add that in western countries the suggestion for most patients is perioperative chemotherapy with FLOT.

(A2) Thank you for the valuable comments. We revised the description of western guidelines according to the reviewer’s comment. We then referred to FLOT as a promising new regimen to prevent recurrence in the discussion. (Lines 42-49 & 286-290)

(C3) You have written the follow up period in days. I would suggest to use months or years rather than days (page 4, line 148)

(A3) Thank you for your suggestion. We realize that patient’s prognosis is usually expressed either with month or year in the clinical practice. On the other hand, expressing the survival time in days allows for more precise calculations when performing survival analysis. We therefore expressed the follow-up period in days. We hope you understand this situation.

(C4) In the section "Conclusions" I think that the conclusion to consider adjuvant chemotherapy in patients with vascular invasion is not a result of your study. You might want to say: "these patients could be consider for participation in future studies of adjuvant chemotherapy".

(A4) Thank you for your comments. We really think so and revised the conclusion accordingly. (Lines 306-309)

Reviewer 3 Report

Nice work. Congratulations. Conclusions part may be improved by removing abbreviations and giving thrust to the study results.

Author Response

(C1) Nice work. Congratulations. Conclusions part may be improved by removing abbreviations and giving thrust to the study results.

(A1) We thank Reviewer 3 for his/her encouraging comment.

We revised the manuscript according to the reviewer’s comment. In order to give thrust to the study results, we exchanged the sentence “VI grade and pT might be potential recurrence predictors” to “VI grade and pT are potential recurrence predictors.” (Lines 303-309)